# HBM4EU Chromates Study: Determinants of Exposure to Hexavalent Chromium in Plating, Welding and Other Occupational Settings

**DOI:** 10.3390/ijerph19063683

**Published:** 2022-03-19

**Authors:** Susana Viegas, Carla Martins, Beatrice Bocca, Radia Bousoumah, Radu Corneliu Duca, Karen S. Galea, Lode Godderis, Ivo Iavicoli, Beata Janasik, Kate Jones, Elizabeth Leese, Veruscka Leso, Sophie Ndaw, An van Nieuwenhuyse, Katrien Poels, Simo P. Porras, Flavia Ruggieri, Maria João Silva, Jelle Verdonck, Wojciech Wasowicz, Paul T. J. Scheepers, Tiina Santonen

**Affiliations:** 1Public Health Research Centre, NOVA National School of Public Health, Universidade NOVA de Lisboa, 1600-560 Lisbon, Portugal; 2Comprehensive Health Research Center (CHRC), 1169-056 Lisbon, Portugal; 3Department of Environment and Health, Istituto Superiore di Sanità, 00161 Rome, Italy; beatrice.bocca@iss.it (B.B.); flavia.ruggieri@iss.it (F.R.); 4French National Research and Safety Institute, 54500 Vandoeuvre-les-Nancy, France; radia.bousoumah@inrs.fr (R.B.); sophie.ndaw@inrs.fr (S.N.); 5Department Health Protection, Laboratoire National de Santé (LNS), 1, Rue Louis Rech, 3555 Dudelange, Luxembourg; radu.duca@lns.etat.lu (R.C.D.); an.vannieuwenhuyse@lns.etat.lu (A.v.N.); 6Centre for Environment and Health, Department of Public Health and Primary Care, KU Leuven (University of Leuven), O&N 5b, Herestraat 49, 3000 Leuven, Belgium; lode.godderis@kuleuven.be (L.G.); katrien.poels@kuleuven.be (K.P.); jelle.verdonck@kuleuven.be (J.V.); 7Institute of Occupational Medicine (IOM), Edinburgh EH14 4AP, UK; karen.galea@iom-world.org; 8IDEWE, External Service for Prevention and Protection at Work, 3001 Heverlee, Belgium; 9Department of Public Health, University of Naples Federico II, 80138 Naples, Italy; ivo.iavicoli@unina.it (I.I.); veruscka.leso@gmail.com (V.L.); 10Nofer Institute of Occupational Medicine, 91-348 Lodz, Poland; beata.janasik@imp.lodz.pl (B.J.); wojciech.wasowicz@imp.lodz.pl (W.W.); 11Health & Safety Executive, Buxton, Derbyshire SK17 9JN, UK; kate.jones@hse.gov.uk (K.J.); liz.leese@hse.gov.uk (E.L.); 12Finnish Institute of Occupational Health, 00250 Helsinki, Finland; simo.porras@ttl.fi (S.P.P.); tiina.santonen@ttl.fi (T.S.); 13Lisbon and ToxOmics—Centre for Toxicogenomics and Human Health, Department of Human Genetics, NOVA Medical School, National Institute of Health Dr. Ricardo Jorge, Universidade NOVA de Lisboa, 1169-056 Lisbon, Portugal; m.joao.silva@insa.min-saude.pt; 14Radboud Institute for Health Sciences, Radboudumc, P.O. Box 9101, 6500 HB Nijmegen, The Netherlands; paul.scheepers@radboudumc.nl

**Keywords:** hexavalent chromium, exposure determinants, risk management measures, occupational hygiene, biomonitoring, air monitoring, dermal exposure

## Abstract

Work-related exposures in industrial processing of chromate (chrome plating, surface treatment and welding) raise concern regarding the health risk of hexavalent chromium (Cr(VI)). In this study, performed under the HBM4EU project, we focused on better understanding the determinants of exposure and recognising how risk management measures (RMMs) contribute to a reduction in exposure. HBM and occupational hygiene data were collected from 399 workers and 203 controls recruited in nine European countries. Urinary total chromium (U-Cr), personal inhalable and respirable dust of Cr and Cr(VI) and Cr from hand wipes were collected. Data on the RMMs were collected by questionnaires. We studied the association between different exposure parameters and the use of RMMs. The relationship between exposure by inhalation and U-Cr in different worker groups was analysed using regression analysis and found a strong association. Automatisation of Cr electroplating dipping explained lower exposure levels in platers. The use of personal protective equipment resulted in lower U-Cr levels in welding, bath plating and painting. An effect of wearing gloves was observed in machining. An effect of local exhaust ventilation and training was observed in welding. Regression analyses showed that in platers, exposure to air level of 5 µg/m^3^ corresponds to U-Cr level of 7 µg/g creatinine. In welders, the same inhalation exposure resulted in lower U-Cr levels reflecting toxicokinetic differences of different chromium species.

## 1. Introduction

Occupational exposure to hexavalent chromium (Cr(VI)) may occur when Cr(VI) compounds are manufactured as end-products (e.g., chromate production) when they are used as starter-products in processes (e.g., electroplating), or when they are formed as process emissions (e.g., in welding) [1]. 

Cr(VI) occupational exposure can occur by inhalation, dermal contact and hand-to-mouth contact [2,3]. Cr(VI) enters cells due to the high membrane permeability to its solubilised forms and is toxic due to its oxidising ability via reactive oxygen species (ROS) produced by the intracellular detoxification process. Moreover, it acts by direct and indirect genotoxic mechanisms, given that Cr(IV) and Cr(V) may form pre-mutagenic DNA- and DNA-protein adducts and that ROS may contribute to DNA single- and double-strand breaks formation, both resulting in genetic instability [4,5]. Several serious adverse health effects have been linked with occupational exposure to Cr(VI) [6]. Cr(VI) is an occupational carcinogen that has been shown to cause lung cancer in humans and has been associated with cancer of the nose and nasal sinuses [7,8].

In the European Union, the use of Cr(VI) compounds (chromates, chromium trioxide and dichromium tris(chromate)) is authorised under the Registration, Evaluation, Authorisation and Restriction of Chemicals (REACH) regulation. REACH was adopted to improve the protection of human health and the environment from the risks that can be posed by chemicals while enhancing the competitiveness of the EU chemicals industry [9]. Process-generated fumes such as those produced during welding operations are not included in REACH. However, occupational safety and health legislation (Occupational Safety and Health framework directive 98/24/EC and Carcinogens and Mutagens Directive (CMD) 2004/37/EC) apply to these types of operations, including substances, mixtures or processes referred to in Annex I of Directive 2004/37/EC meeting criteria [10].

The current binding occupational exposure limit value (BOELV) adopted under EU Directive 2004/37/EC is 0.010 mg Cr(VI)/m^3^ until 17 January 2025. After that date, a reduced limit of 0.005 mg Cr(VI)/m^3^ will be applied. In the case of welding, plasma-cutting activities or similar processes that generate fumes, there is a derogation with an occupational exposure limit (OEL) value of 0.025 mg Cr(VI)/m^3^, also until 17 January 2025. After the transposition date, the limit will also be 0.005 mg Cr(VI)/m^3^ for these welding or plasma-cutting processes [10]. In France and The Netherlands, OELs of 0.001 mg/m^3^ have already been set for Cr(VI) [11,12], and these are currently the most rigorous for Cr(VI) in the EU.

In a recently published systematic review of biomonitoring data on occupational exposure to Cr(VI) [13], it was concluded that improved working conditions, efficient use of personal protective equipment (PPE), better exposure control and increased risk awareness could contribute effectively to Cr exposure reduction. Verdonck et al. identified a need to further investigate the contribution of the different exposure routes (mainly inhalation and ingestion due to hand-to-mouth contact) in the different occupational settings to allow better guidance on which control measures should be prioritised in each setting [13]. Verdonck et al. also highlighted that some specific tasks are associated with high exposure levels, such as metal processing and finishing and welding [13].

The commonly used biomarker to assess Cr exposure at occupational settings is the urinary total Cr (U-Cr), for which several biological limit values (BLV) are available in EU countries such as France and Finland [12,14]. However, this total Cr is the sum of different oxidation states and, therefore, not specific for Cr(VI).

The EU human biomonitoring initiative (HBM4EU) is a Joint Programme aiming to standardise and use biomonitoring to understand human exposure to chemicals (via the environment, in occupational settings or through using consumer products) and related health risks, with the aim to improve chemical risk assessment and management as well as supporting policymaking [15]. HBM4EU is a joint effort of 30 countries, the European Environment Agency and the European Commission, co-funded under Horizon 2020 (www.hbm4eu.eu (accessed on 12 December 2021)).

In the scope of the HBM4EU, the Chromates Study was developed [16] with the main aim of providing EU-relevant data on Cr(VI) internal exposure and early biological effects in occupational settings. These data should be used as scientific evidence for regulatory risk assessment and decision-making under EU chemical legislation and occupational safety and health legislation.

Santonen et al. provided details of the overall results and recommendations from the HBM4EU Chromates Study for the biomonitoring of occupational exposure to Cr(VI) [17]. It was concluded that U-Cr continued to be a valuable biomarker as a first approach for the assessment of total Cr internal exposure, and high correlations were observed between U-Cr levels and both air Cr(VI) and dermal total Cr exposure [17]. The authors reported that the highest internal exposures were observed in the use of Cr(VI) in bath plating. The use of respiratory protection equipment (RPE) contributed to a reduction in U-Cr in paint applications. Not all chrome plating workers used RPE, or this use was restricted to specific tasks of short duration, such as collecting samples from Cr baths [17]. It was hypothesised that less frequent use of RPE among bath platers might explain the higher internal exposure of this sub-category when compared to, e.g., welders, which showed very similar (or even slightly higher) inhalable air levels of Cr(VI) [17].

It is useful to further analyse our data for a better understanding of the determinants of exposure in each of the industrial sectors studied and identify additional risk management measures (RMMs) both at a company and regulatory level. Therefore, this manuscript aims to provide a more in-depth analysis and assessment of exposure by use of urinary biomarkers, occupational hygiene samples and by use of contextual information with the aim to identify exposure determinants that contribute to Cr exposure and evaluate the influence of implemented RMMs exposure control.

## 2. Materials and Methods

### 2.1. Study Population and Recruitment

Nine countries participated in the HBM4EU Chromates study (Belgium, Finland, France, Italy, Luxembourg, Poland, Portugal, The Netherlands, and the United Kingdom), all applying harmonised methods and following the same Standard Operating Procedures (SOPs) developed in the scope of the HBM4EU Chromates study [16].

The study population consisted of workers of companies with activities that are known to be associated with occupational exposure to Cr(VI), such as (i) chrome plating, (ii) surface treatment by sanding, spraying or painting, and (iii) stainless-steel welding. After the sampling campaigns and detailed characterisation of the companies and exposure scenarios involved in the study, it was possible to divide the workers according to the specific activities in which they were involved, namely welding, bath plating, painting, machining, steel production, thermal spraying and maintenance and laboratory work. This division was performed based on the workers’ tasks, their frequencies, as well the materials and processes applied.

Recruitment of the companies and workers; collection and analysis of biomonitoring samples (urine) and industrial hygiene samples (air and hand wipe); and collection of contextual information was conducted using two questionnaires [16,17], and are therefore only briefly described in the following sections.

Study protocols were submitted for approval by ethics boards in each of the participating countries, with the approvals being granted before the study participants were enrolled [16,17,18].

This study enrolled 602 individuals (399 workers and 203 controls) from the nine countries involved. In this paper, we focused on the workers’ exposure (U-Cr, air monitoring and hand wipe samples results) and contextual data collected using two questionnaires (one filled by company representative and the other by the workers). The controls used in the statistical analysis performed in this study were the same as used in the paper of Santonen et al. (2022), where the overall results of the HBM4EU Chromates study were described. In brief, controls were unexposed workers recruited either within the same company but from activities that are known not to be associated with Cr(VI) exposure (for example, office staff) (“within company controls”) or from other companies with no activities associated with Cr(VI) exposure (“outwith company controls”) [17].

### 2.2. Sampling

All the samples were collected between October 2018 and December 2020, following the procedures defined in dedicated SOPs [16]. The samples analyses were performed as described in previous publications [16,17].

#### 2.2.1. Air Samples

Personal inhalable dust was sampled in the breathing zone using an IOM sampling head (flow rate 2 L/min), whereas the respirable dust fraction was collected using the Higgins Dewell type (or similar) cyclone sampling heads. The flow rates followed were the ones recommended by the samplers’ manufacturers. The samplers were placed in the breathing zone of workers. The inhalable and respirable sampling head cassettes were loaded with pre-weighed 25 mm PVC-filters (GLA-5000, 5 µm pore size). In the case of welders, alternatively, the SKC Mini-sampler was used, loaded with a pre-weighed 13 mm MCE filter, at a flow rate of 0.75 L/min, placed under the welding visor. The SKC mini-samplers were used only in the UK, Belgium and Luxembourg for the collection of total Cr. Moreover, the few Cr(VI) samples collected under the welding RPE were collected using SKC mini-samplers. All of these sampling devices adhere to CEN-EN 481:1993 Workplace atmospheres—size fraction definitions for measurement of airborne particles. The inhalable samples were collected for a representative period of the work shift (>75%) [17]. The air samples were first analysed gravimetrically and subsequently for total Cr and Cr(VI) by OSHA Method ID-125G [19] and ISO 16740 Method [20], respectively, with some minor adaptations performed by some laboratories. Not all countries analysed their air samples for both total Cr and Cr(VI), and only four countries (Finland, France, Italy and Poland) provided data on both total Cr and Cr(VI). Belgium, Luxembourg and United Kingdom measured total Cr, and The Netherlands and Portugal measured only Cr(VI). All the air samples results were provided as 8 h TWA.

#### 2.2.2. Dermal Wipe Samples

Dermal wipe samples (hand wipes) were collected from both hands using SKC Ghost sampling wipes [19,21] or similar. Samples were collected from both hands at specific periods during the working shift (pre-shift, first break period, lunch, and post-shift), using a standardised wiping procedure [16]. The wipes were analysed for total Cr using OSHA Method ID-125G [19]. Average hand areas of 535 cm^2^ per male hand (total 1070 cm^2^ for both hands) and 445 cm^2^ per female hand (total 890 cm^2^ for both hands) [22] were used in subsequent calculations [16,17]. The number of wipe samples collected per worker was dependent on the duration of each task involving exposure to Cr and the number of breaks/hand washings during the shift, ranging from 2 to 6 per worker.

For both the inhalation and dermal samples, an appropriate number of field blanks samples were also collected and analysed. These samples were labelled as field samples and sent along with the field samples for laboratory analysis. These blank field samples are used to mitigate the potential for unrecognised contamination due to media or sample handling during the field work [21].

#### 2.2.3. Urine Samples

Urine samples were collected at the beginning (pre-shift) and the end (post-shift) of the work shift at the end of the working week (on Thursday or Friday). Containers for sample collection were previously decontaminated (10% HNO_3_) to avoid background contamination. After collection, urine samples were homogenised and aliquoted in several pre-labelled tubes and stored at −20 °C. Urinary creatinine concentrations were measured, and U-Cr results were adjusted to creatinine (µg Cr/g creatinine) accordingly [23,24].

### 2.3. Contextual Data Collected

Two specific questionnaires were developed and used to collect relevant contextual information. The first one was a self-administered questionnaire completed by a company representative prior to the sampling campaign (questionnaire 1). The second one was an interviewer-led post-shift worker questionnaire (questionnaire 2) completed while interviewing the worker as close as possible to the end of the work shift [17].

The company questionnaire aimed to collect general information on the company. Details regarding previous training on safety issues related to the working tasks, previous exposure monitoring campaigns, and occupational health and safety practices were obtained. Details of the general operating conditions related to chrome plating, surface treatment and welding operations (as applicable) were also collected [16].

The worker questionnaire was more detailed. Different questions were prepared and administered to the subgroups involved in different activities included in the HBM4EU Chromates study (i.e., Cr plating in baths, welders, and surface treatment workers). A detailed description of the tasks performed on the sampling day was collected. In addition, details of the RMMs were collected, e.g., presence of local exhaust ventilation (LEV), availability and use of PPE, previous information and training on safety issues, the possibility of washing the hands during work, the existence of a dedicated place for storage of working clothes and RPE. Possible background exposures from non-workplace sources (e.g., hobbies, diet, air pollution based on home location) were also investigated in this questionnaire [16]. However, not all the collected information could provide data suitable to perform a detailed statistical analysis due to missing answers for various reasons (e.g., workers without availability to answer the questionnaire). Therefore, it was only possible to use the information to investigate the influence on workers’ exposure results for some variables. Determinants of exposure included in the analysis are presented in Table 1.

### 2.4. Data Management and Statistical Analysis

A harmonised Microsoft Excel (Microsoft, Redmond, WA, USA) data template was prepared for use by all the research teams involved in the study to allow the pooling of the data for analysis. The dataset included both contextual data (questionnaire data) and the results from samples analyses (biomonitoring and industrial hygiene samples). After minor spreadsheet calculations and data editing cleaning, the final data template was imported into IBM© SPSS© Statistics software (IBM Corp., Armonk, NY, USA, Released 2019. IBM SPSS Statistics for Windows, Version 26.0. Armonk, NY: IBM Corp.) for statistical analysis.

The results of total U-Cr were presented as creatinine-adjusted concentrations (μg/g creatinine). Regarding the data treatment of non-detects (<limit of quantification (LOQ)), the substitution by a fixed value was used considering a middle-bound approach (<LOQ = ½ LOQ) [25]. This approach was already used in the previous paper describing the overall results [17]. Descriptive statistics were performed for quantitative variables (mean, median, and percentiles 75 (P75) and 95 (P95)) and qualitative variables (frequencies). The normality of distributions was checked with the Shapiro–Wilk test. Since not all were normally distributed, non-parametric tests were used for further statistical analysis. The correlation among continuous variables was determined with the Spearman correlation coefficient (r_s_) (r_s_ ≤0.2 = poor; 0.2 < r_s_ ≤ 0.5 = fair; 0.5 < r_s_ ≤ 0.7 = moderate; 0.7 < r_s_ ≤ 1.0 = very strong) [26]. For the work-related variables, two levels of aggregation were considered for the statistical analysis: the first level for occupational setting (welding, Cr plating and surface treatment) and the second level for several tasks within each setting (e.g., readjustment of the electrolyte, spraying in spray cabin/spray booth). Variables were recoded considering these two levels of aggregation whenever needed. In order to ensure an adequate number of data points, the RMMs were dichotomised for inclusion in the statistical analysis as follows: use of RPE (y/n), daily fit check of RPE (y/n), use of gloves (y/n), the existence of a dedicated place for storing working clothes (y/n), the existence of a dedicated place for storing RPE (y/n), presence of LEV (y/n), previous training in OSH issues (y/n) and previous monitoring campaigns (environmental, human biomonitoring, both, none).

The concentrations of total Cr and Cr(VI) in different samples were analysed (when suitable) regarding statistically significant differences between (*i*) urine samples pre-shift and urine samples post-shift (two related samples, Wilcoxon test) and (*ii*) self-reported activities reported by workers and controls (independent samples, Mann–Whitney test, and Kruskal–Wallis test).

The influence of the RMMs on Cr in post-shift urine samples and industrial hygiene samples (air and hand wipe) was assessed through Mann–Whitney test or Kruskal–Wallis test considering the number of categories in each variable. A level of significance of 5% was considered for all the analysis. Due to the low number of observations in some activities, these analyses were only carried out for welding, bath plating, painting and machining activities.

Linear regression modelling with a single explanatory variable was performed or post-shift U-Cr (µg/g creatinine) with inhalable Cr(VI) outside RPE (µg/m^3^). The model was run for all workers and separately for bath platers and welders; either all platers and welders combined or those not using RPE in their tasks were analysed separately.

## 3. Results

### 3.1. Companies Involved in the Study

The general characteristics of the company and workers’ activity, as well as the operational conditions, are presented in Appendix A.

Approximately 40% of workers were from metallurgy, and 15.0% of workers were from steel and steel products/metals sectors. The workers were categorised into seven types of activities that were primarily targeted in the recruitment and used in the subsequent questionnaires for companies and workers (e.g., Cr plating in baths, surface treatment, stainless-steel welders). Almost half of the workers (48.9%) were employed in the welding activities, followed by chromate plating (22.6%) and painting (13.0%).

The following working conditions were reported: indoor environment (96%), 8 h shift duration (78.9%) and fixed day shifts (60.9%). Ten companies, corresponding to 68 workers (17.0%), did not report any previous environmental (air and/or dermal) monitoring and/or biomonitoring campaigns. Of the workers, 70.9% were employed in companies that have previously developed environmental monitoring and human biomonitoring campaigns. Concerning previous training in OSH issues, most of the workers (83.7%) were engaged in companies where training was performed. The vast majority of the workers stated that it was possible to wash their hands during the work shift (97.5%), and there was a dedicated place for storing the working clothes (83.2%) and the RPE (62.9%).

### 3.2. Study Population

The main characteristics of the studied population are summarised in Appendix A. As already reported [17], exposed workers were mainly men (97.7%), and the age distribution was 42 ± 11 years old (mean ± SD). Considering the distribution by country of the enrolled participants, it ranged between 4.0% (Luxembourg) and 17.7% (Belgium). The participating countries recruited workers from the three activity sectors, except for Luxembourg, which enrolled only welders, and Poland, which recruited mainly welders. The Netherlands studied only bath plating workers, and the United Kingdom recruited welding, bath plating, machining, and maintenance workers. Concerning the location of their homes, workers lived mainly in urban areas (63.7%), with most of the workers reporting a low density of traffic (52.1%). Regarding smoking status, a total of 140 (35.6%) workers were smokers, and 155 (39.4%) workers were non-smokers.

Regarding previous work experience, welders, bath platers, painters and machining workers reported years of work in welding, metal plating, painting, or spraying. Thermal spraying, maintenance and laboratory workers reported a work history in metal plating and in other jobs in the metal industry (Appendix A).

Mean length of experience in their jobs varied between 1.5 years for painting or spraying and also bath plating (with a maximum of 42 years and 46 years, respectively), 6.5 for welding (the maximum being 47 years) and 9.5 years for other metal works (e.g., machining, fitter, cutter, grinder, with a maximum of 39 years) (Appendix A). The self-reported number of years worked was positively and significantly correlated with pre- and post-shift levels of U-Cr only for welders (n = 162, r_s_ = 0.191, *p* = 0.015 and n = 158, r_s_ = 0.168, *p* = 0.034, respectively).

### 3.3. Total Cr and Cr(VI) in Industrial Hygiene Samples (Air and Wipes)

Table 2 presents the results of total Cr and Cr(VI) in air samples and total Cr in hand wipes considering the different activities. The Cr(VI) air monitoring results (but not total Cr results) were earlier reported by Santonen and co-workers (2022).

Concerning the inhalable air samples, in the results outside RPE, thermal spraying activity presented the highest values of total Cr, with statistically significant differences from steel production workers (*p* = 0.002) and bath platers (*p* = 0.001). These differences were also found between welders and bath platers (*p* = 0.025). Regarding Cr(VI) levels, statistically significant differences were found across all chrome processing activities (*p* < 0.001) and mainly driven by the following five pairs: thermal spraying activity in relation to machining (*p* < 0.001), bath plating (*p* = 0.007) and welding activities (*p* = 0.024); painting in relation to machining (*p* < 0.001) and bath plating (*p* = 0.007); and machining in relation to welding (*p* = 0.005).

For the welding sector, total Cr values measured inside the RPE were also provided (n = 34) and showed that exposure to Cr still occurs even when RPE is used (GM of 3.7 µg/m^3^). Inhalable Cr(VI) levels inside RPE were measured only from a few welders with a GM of 1.0 µg/m^3^. Concerning the respirable fraction, the welding and thermal spraying activities presented the highest values; however, no statistically significant differences were found.

Regarding wipe samples, a minimum of two and a maximum of six hand wipes were collected per worker during a working day, adding to a total of 267 hand wipes collected within the nine countries. The highest values were obtained for thermal spraying (P95 = 46.6 µg/cm^2^), with statistically significant differences from the remaining activities.

### 3.4. Total Cr in Urine

U-Cr concentrations measured from the different activity sectors are presented in Table 3 as well as correlation analysis (r_s_) between results of U-Cr of pre-shift and post-shift samples.

As already reported [17], when considering the total group of the exposed workers, pre-shift levels were significantly higher compared to the levels observed within company controls (*p* < 0.001), outwith company controls (*p* = 0.025) and all controls (*p* < 0.001). Additionally, as compared to the pre-shift levels, all worker groups showed statistically significant increased post-shift levels (*p* < 0.001). If considering the controls, within company controls presented statistically significant differences from outwith company controls regarding total U-Cr levels (*p* = 0.002). The correlation U-Cr levels between pre-shift and post-shift urine samples was moderate for machining (r_s_ = 0.588) and very strong for welding (r_s_ = 0.797), bath plating (r_s_ = 0.892) and painting (r_s_ = 0.703) (Table 3). Regarding pre-shift U-Cr levels, statistically significant differences were found between workers and the different groups of controls. Post-shift workers’ levels of U-Cr were compared with controls’ levels of U-Cr, and statistically significant differences were found among the four activities and the two groups of controls. When considering the results by activity, GM results of post-shift U-Cr concentrations are similar between the different activities, with bath plating and machining presenting the higher values for the P95 of exposure and welders presenting statistically significant differences from bath platers (*p* = 0.030).

When analysing chrome plating activities more with more detail, in chrome electroplating dipping (includes loading the tanks), significant differences in post-shift U-Cr between automatic and manual processes were observed (*p* = 0.037) (Figure 1, Appendix A).

Concerning welding activities, the most common welding processes reported were tungsten inert gas (TIG) (39.5%) and shielded metal arc welding (SMAW) (17.4%). No differences were observed in U-Cr levels between these welding techniques.

### 3.5. Correlations between Different Exposure Metrics

The results of bivariate analysis and considering all workers’ samples (total Cr in urine and wipes, total Cr and Cr(VI) in air) are presented in Table 4.

Spearman coefficients showed very strong correlations between total U-Cr levels of pre- and post-shift samples (r_s_ = 0.795) (Table 5). Very strong correlations were also observed between levels of total Cr and Cr(VI) in inhalation fraction outside the RPE and the respirable fraction outside the RPE (r_s_ = 0.800 and r_s_ = 0.791, respectively). Moderate correlations were observed for levels of total Cr in post-shift urine and respirable dust Cr(VI) outside RPE (r_s_ = 0.694), inhalable dust Cr(VI) inside RPE (r_s_ = 0.514), levels of total Cr and Cr (VI) in inhalable dust outside RPE (r_s_ = 0.609), levels of total Cr in inhalable dust outside RPE and hand wipe (r_s_ = 0.606) and levels of Cr(VI) in respirable dust outside RPE and hand wipe (r_s_ = 0.639).

In addition, when considering different work activities (bath plating, welding and painting), correlations were found between levels of total Cr in post-shift urine samples and levels of Cr(VI) in inhalable air outside RPE samples: r_s_ values of 0.783, 0.592 and 0.821, for bath plating, welding and painting, respectively (data not shown). However, it should be noted that for painting, the number of air measurements was only 7, whereas for bath plating and welding 57 and 107 measurements were considered, respectively.

Regression analyses were performed to study the relationship between inhalable or respirable Cr(VI) levels (outside RPE) and U-Cr levels. Correlations between inhalable or respirable Cr(VI) and U-Cr levels were only moderate when all the workers were combined (Table 4). Therefore, separate analyses were made for chrome platers and for welders. In the case of welders, regression analyses were made only between inhalable Cr(VI) and U-Cr due to the low number of respirable Cr(VI) measurements. When platers were analysed separately, a Spearman correlation coefficient between inhalable Cr(VI) and U-Cr was improved to r_s_ = 0.783, and a regression equation of y = 1.174 + 0.745x was obtained (n = 57, including platers with both U-Cr and inhalable Cr(VI) measurements). However, goodness-of-fit (R^2^) remained at a relatively low 0.349. Therefore, regression analyses were run separately for those chrome platers that had not worn RPE during the day of sampling (n = 42, including platers with both U-Cr and inhalable Cr(VI) measurements). The regression equation obtained was y = 0.742 + 1.235x with a R^2^ = 0.679 and a Spearman correlation coefficient (r_s_ = 0.858); this is presented in Figure 2a. In the case of respirable Cr(VI), regression equation of Y = 1.289 + 1.989x was obtained for all platers, with R^2^ = 0.410 and a Spearman correlation coefficient (r_s_ = 0.805). Separate analyses of workers not using RPE did not improve R^2^. It should be noted that the highest respirable air level for platers was 3.1 µg/m^3^, which may make extrapolation to higher air levels uncertain.

Similar analyses were also performed for welders. When all welders were analysed (n = 106, including all welders with both U-Cr and inhalable Cr(VI) measurements), a moderate Spearman correlation coefficient of 0.429 was obtained, but R^2^ for linear regression was only 0.049. When those welders who had not used RPE during the day of sampling were analysed separately, a regression equation of y = 0.647 + 0.541x was obtained, with a moderate Spearman correlation coefficient (r_s_ = 0.515) and R^2^ was improved as 0.324. This regression analysis is presented in Figure 2b. Regression analyses were also run for all data and separately for platers and welders, using either log-transformed data or a non-linear regression model (quadratic equation). These did not, however, result in any significant improvement in fits (data not shown).

### 3.6. Determinants of Exposure

The RMMs in place in the companies enrolled in the study and other contextual information collected by the questionnaires are presented in Table 5, as well as their impact on levels of total Cr and Cr(VI) in urine and in industrial hygiene samples (air and wipe), when applicable. The detailed characteristics of the RMMs and other variables by each activity are fully described and presented in Appendix A.

The use of the RPE had an influence on total U-Cr levels for workers in welding, bath plating and painting activities. Workers using RPE presented significantly lower levels of total U-Cr. This effect was observed in all worker subgroups except for machining. The presence of LEV had an influence on levels of total Cr in urine and in the inhalable dust fractions outside RPE in welding. This influence was also observed in the results of hand wipes from painting. In the case of welding, the levels of total Cr and Cr(VI) were significantly lower in the presence of LEV. Moreover, for welding, workers who had received previous training in OSH issues presented significantly lower total Cr levels in urine and hand wipes, thus emphasising the importance of instruction and training. However, companies that reported on previous monitoring surveys, including the parallel assessment of the workplace and human biomonitoring assessment, had significantly higher levels of total Cr in urine samples and of Cr(VI) in inhalable air samples (outside and inside the RPE), as compared to companies that only had relied on industrial hygiene measurements previously. An opposite trend was observed in bath plating companies, where previous monitoring actions resulted in significantly lower levels in most of the exposure metrics used.

In bath plating, when a dedicated place for storing work clothes is present, significantly higher levels of total U-Cr and total Cr in hand wipes were observed. On the contrary, when a dedicated place for storing RPE was available, significantly lower levels of total U-Cr and total Cr in hand wipes were detected. In painting, a similar trend was observed; when a dedicated place for storing work clothes was present, workers showed statistically significant higher levels of total U-Cr. However, these data should be interpreted with some caution since a low number of reports were registered, particularly for the “no” option (Yes = 31, No = 7).

The statistical analysis did not reveal any association between the RMM and the levels of total Cr in inhalable inside RPE and respirable outside RPE air samples (data not shown).

Concerning the non-workplace exposure sources (Table 1), the statistical tests showed no effect on exposure metrics (Appendix A). We only observed an influence in exposure related to their home location in the subgroup of bath platers, where workers living in urban areas presented significantly higher levels in total U-Cr levels (*p* = 0.026 and *p* = 0.010 for post-shift and pre-shift total U-Cr, respectively) (Appendix A).

## 4. Discussion

This study aimed to identify the exposure determinants that result in increased Cr exposure and the RMMs contributing to control the exposure using exposure biomarkers (U-Cr), industrial hygiene data (air and hand wipe samples) and contextual data collected. This was accomplished since it was possible to recognise some of the RMMs that impacted workers’ exposure to the activities where exposure to Cr(VI) can occur. The information obtained by the major activities (welding, chromate plating and surface treatment, including painting and machining) provides scientific support for informed decisions made by regulatory and policy agencies involving stakeholders and policy actors. This is possible even when different regulatory frameworks are in place, such as the OSH legislation, including CMD that applies for welding and, for all the other activities, besides OSH directives, REACH regulation is also applicable. Therefore, as long as substitution is not possible, knowing which RMMs are more effective allows recommendations, investments and actions focused on reducing workers’ exposure as envisaged by the different regulatory frameworks in place.

### 4.1. Workers Exposure

The increase in U-Cr levels of workers when comparing pre-shift samples collected before the start of the workweek (pre-shift) with samples collected at the end of the shift (post-shift) towards the end of the workweek indicated work-related uptake of chrome over the preceding workweek. This is most probably related to the tasks performed in the workplace. Indeed, environmental exposure to Cr(VI) is known to occur mainly through tobacco smoke, inhalation of polluted air or ingestion of contaminated water in individuals living near industrial or contaminated areas [7]. It was only possible to observe an influence in Cr exposure related to the home location in the case of bath plating. No influence of tobacco smoke was detected in exposed workers or in controls. Therefore, it is possible to state that the workplace environment is the predominant contributor to the total body burden of Cr observed in this study.

When considering the results with respect to the activity performed, mean and median results were similar between the different activities, although bath plating and steel production workers presented the higher values for the P95 of exposure. In the publication by Santonen and co-workers [17], the bath plating workers group was reported to have higher exposures, and we suggested that this might be related to most of the plating workers not wearing RPE. This was confirmed by our analyses of contextual data and U-Cr, showing that more than half of the workers (54.4%) reported not using RPE (and 17% did not provide information on RPE use) and that the use of RPE was associated with lower Cr exposure (Table 4). Moreover, in chrome plating, several tasks are still performed manually, such as baths readjustment (72.7%) and Cr electroplating dipping (79.8%), also contributing to the exposure found. It was possible to confirm the influence observed in U-Cr levels.

Additionally, pre-shift levels of the total group of workers were significantly higher when compared to the levels observed in the controls indicating retention of Cr in the body [17,27]. Furthermore, a more detailed analysis showed a moderate correlation in U-Cr levels between pre-shift and post-shift urine samples from machining workers and a very strong correlation for welding, bath plating and painting workers (Table 3). These moderate to very strong associations indicate a slow excretion rate of at least some of the inhaled Cr species [28]. Previously, Scheepers et al. and Pesch et al. already reported a strong correlation between pre- and post-shift U-Cr levels in welders, indicating a slow excretion rate [28,29]. Therefore, besides considering the kinetics and bioavailability of the Cr species present in also need to consider that when using post-shift samples, soluble Cr species are more excreted into the urine than more insoluble Cr species [28].

This study focused mainly on biomonitoring but also included the collection of industrial hygiene samples (air and hand wipes) to support the interpretation of study results [17,18]. Very strong correlations were observed between inhalation and respirable fractions levels of total Cr and Cr (VI); outside the RPE is of great importance, showing that task-based activities that increase total Cr exposure may also increase the exposure to Cr(VI) (Table 4). Therefore, even when it is not possible to measure Cr(VI), the more generally used Cr measurement in air provides an insight into whether exposure to Cr(VI) is occurring and how it is evolving. In the case of welding, previous surveys [30] demonstrated that the welding method influenced the relationship between Cr(VI) and total Cr. The reported proportion of Cr found in fumes from manual metal arc (MMA) welding identified as Cr(VI) varied significantly (15–88% of total Cr), while fractions were smaller for tungsten inert gas (TIG) and metal active gas (MAG) welding (7–40% and 2–17%, respectively). In our study, TIG was the welding process most reported by companies (39.5%), and this might influence the results obtained.

We observed that the highest P95 values for the inhalable fraction for total Cr were obtained in thermal spraying, welding, and bath plating. In the case of Cr(VI), the highest P95 values were for painting, thermal spraying (only five workers from the same company) and bath plating. The same trend was observed for the total Cr in the respirable fraction, where welding, thermal spraying and bath plating presented the higher values for P95. In the case of Cr(VI), welding and bath plating were the activities with higher P95 for the respirable fraction. Although welders and bath plating workers showed very similar inhalable and respirable Cr(VI) levels, the fact that only 29% of platers claimed to use RPE might contribute to explaining the higher U-Cr P95 value obtained for bath plating workers. Although thermal spraying obtained high levels in the industrial hygiene samples (air and hand wipe), this did not result in high U-Cr, probably because most of the tasks were performed with RPE and only in occasional maintenance activities workers did not report the use of RPE (Appendix A).

What is interesting to note is that in plating, we observed significant exposure to trivalent chromium (Cr(III)) compounds. In welding activities, it is well known that both Cr(III) and Cr(VI) species are formed in the process, and Cr(III) exposure may contribute to the total body burden of Cr regardless of its much poorer absorption when compared to Cr(VI). Although the P95 levels of total Cr remained below the OEL of 0.5 mg/m^3^ commonly applied for Cr(III) in Europe, it is noted that the American Conference of Governmental Industrial Hygienists recently reduced their TLV for Cr(III) to 0.003 mg/m^3^ on the basis of Cr(III) non-cancer lung effects [31]. In this study, most GM levels observed for total Cr are higher, including most Cr(III) exposures in the majority of the tasks evaluated in this study.

When comparing our observed Cr(VI) air concentrations with the BOEL set under EU Directive 2004/37/EC of 10 μg/m^3^ (8 h time-weighted average (8 h TWA)), we can conclude that for painting (n = 11) and thermal spraying (n = 5), in the inhalable fraction, the mean and P75 and P95 values exceeded this provisional limit value of 10 μg/m^3^. After the introduction of a stricter OEL (8 h TWA) of 5 μg/m^3^ by 17 January 2025, P95 value of the inhalable fraction would also be exceed in the case of bath plating. These overall results suggest that there is a need for more effective RMMs, particularly in the case of bath plating, painting, and thermal spraying, to achieve lower exposure levels.

As previously reported [17] for welding, Cr(VI) levels are below the proposed BOELV of 5 μg/m^3^. However, the Cr and Cr(VI) levels inside the RPE indicated that exposure to Cr(VI) still occurs (mean value of 1.6 μg/m^3^ and P95 of 4.1 μg/m^3^, with values ranging from 0.1 to 44.3 μg/m^3^), although below the OEL in most of the cases. Indeed, RPE has an important role in controlling exposure, particularly if in combination with other RMMs such as LEV [32,33].

Although the BOEL for Cr(VI) is set for the inhalable dust fraction, we should also reflect on the findings obtained for the respirable fraction because these particles penetrate deeply into the airways and are slowly cleared. Only three air samples of the respirable fraction for welding exceeded the stricter limit value of 5 μg/m^3^.

The correlations found between Cr in inhalable air and Cr(VI) in respirable air levels with the total Cr in the wipes are also relevant, suggesting that air contamination contributes to the surfaces and hand contamination. Besides exposure by inhalation, air contaminants may also indirectly contribute to Cr exposure by ingestion. Thermal spraying (only five workers, see [17]), bath plating and welding had the highest P95 values for Cr in hand wipes. We did not observe any significant exposures in painting, probably because workers use gloves more frequently due to direct contact with paints and solvents, thus likely more effectively preventing hands from becoming contaminated. In machining, hand wipe results were among the highest P95 values. This can be explained by the more frequent manual handling of the surfaces, with only 29% of the workers reporting the use of gloves. Additionally, the fact that most of the machining workers (71%) reported that LEV was not available in the workplaces might also contribute to explaining the results found.

### 4.2. Relevance of U-Cr and BM for the Exposure Assessment to Cr(VI)

We found moderate correlations between U-Cr in post-shift samples with inhalable and respirable air of Cr(VI), total Cr in inhalable air and also with total Cr in hand wipe samples. These correlations support the added value of using U-Cr as a primary method for the biomonitoring of Cr(VI) exposure at workplaces [17]. Additionally, the very strong correlations between total U-Cr levels of pre- and post-shift samples (r_s_ = 0.795) suggest that U-Cr may reflect historic exposure in addition to recent exposure. This may explain why differences between manual and automatic Cr electroplating dipping were observed based on differences in U-Cr levels only.

In this study, HBM indicated ingestion contributed to explaining overall exposure in bath plating workers. This suggestion is based on finding comparatively the highest values for the P95 concerning hand wipe contamination and the higher values of U-Cr for the P95 of exposure in chrome platers, even when considering the air sample levels that were not amongst the highest compared to other types of chromate processing. Positive moderate correlations between levels of total Cr in post-shift urine and hand wipe samples (r_s_ = 0.606) confirm that ingestion due to dermal contamination is significant in terms of contribution to the overall systemic dose. A similar conclusion was reported in previous biomonitoring studies in the electroplating industry [2,34]. These findings underpin the potential added value of HBM as only exposure biomarkers integrate the contribution from all exposure routes [3,35].

We also performed regression analyses to explore the relationship between U-Cr and inhalable or respirable Cr(VI) levels in air samples. Good correlations were seen for platers, especially when those platers’ who did not use RPE were analysed separately. The obtained regression equation can be used to set BLVs corresponding to OELs (set for inhalable fraction). The regression analysis published by Lindberg and Vesterberg [36] was already used as a basis for deriving a BLV for Cr(VI) in bath plating. Another widely used regression equation for Cr(VI) in electroplating was published by Chen et al. [37]. Our regression analysis made for bath plating workers supports the analysis by Chen et al. [37]. By using Chen et al.’s regression equation, OEL of 5 µg/m^3^ can be calculated to correspond to U-Cr levels of 8.8 µg/g crea [37]. This is close to the value of 7 µg/g creatinine (~9.5 µg/L if an average creatinine excretion of 1.36 g/L is used [24]) obtained using our regression equation for platers (y = 0.742 + 1.235x). However, if regression equation by Lindberg and Vesterberg [36] is used to set a BLV corresponding to the OEL of 5 µg/m^3^, a value of 350 nmol/L (i.e., 18 µg/L or ~13 µg/g creatinine, if an average creatinine excretion of 1.36 g/L is used) is obtained. Lindberg and Vesterberg’s paper did not report the goodness of fit (R^2^), but correlation coefficients were somewhat lower than in our study [36]. The reason for higher levels in Lindberg and Vesterberg [36] cannot be stated for sure, but it might be related to higher surface and hand contamination (contributing to internal levels in Cr(VI)) in electroplating companies at the end of the 1970′s to the beginning of 1980′s than currently. We suggest that Lindberg and Vesterberg’s [36] regression equation should not continue to be used for BLV setting for Cr(VI).

The regression equation obtained for respirable Cr(VI) in chrome platers gives slightly higher U-Cr levels for the same air Cr(VI) levels (e.g., 5 µg/m^3^ corresponding to about U-Cr levels of 11 µg/g creatinine), which could be hypothesised to be due to better absorption of smaller particles. However, it should be noted that goodness of fit is poorer when compared to regression curves made for inhalable fraction, and the highest respirable air level for platers was 3.1 µg/m^3^, making this relationship more uncertain, especially at higher air levels. In addition, the current OELs were set for inhalable fractions, making the relationship between inhalable Cr(VI) and U-Cr more relevant for the BLV setting.

We also observed a moderate correlation between U-Cr and air inhalable Cr(VI) in welders who were not using RPE. The regression curve for welders differs from the curve observed for platers: similar air levels resulted in almost two-times higher U-Cr levels in platers compared to the welders. For example, in welders, OEL of 5 µg/m^3^ corresponded approximately urinary Cr levels of 3.4 µg/g creatinine. The difference between bath plating workers and welders is not surprising since platers are exposed to highly water-soluble Cr(VI) compounds from chromic acid aerosols, whereas welders are exposed to Cr(VI) oxides from the particle phase of welding fumes. Differences in the type of Cr(VI) emissions, including water-solubility of the Cr(VI) compounds and size of the particles, are likely to affect the toxicokinetic of Cr and resulting urinary levels [38]. Although the correlation between air and urinary levels in welders was only moderate and goodness-of-fit (R2) was relatively low, this is an important finding since it shows that in welders, it is not possible to exclude exceedance of Cr(VI) OEL of 5 µg/m^3^ even if urinary Cr levels would stay below 5 µg/g creatinine. To our knowledge, this is the first study reporting regression equations for welders at relatively low exposure levels reflecting current work practices.

### 4.3. Exposure Determinants

#### 4.3.1. Use of RPE and Daily Fit Check 

In chrome plating, the infrequent use of RPE (only 29% of the workers reported to use RPE) influences workers’ exposure, as seen in Table 4 by a statistically significant effect of RPE use on U-Cr levels. A similar effect was observed for RPE use by welders and painting workers. These effects were not seen in machining.

These analyses further confirm our earlier assumption that less frequent use of RPE among Cr plating workers when compared with other sectors—painting 76% and welders 64%—may be one factor explaining the higher internal exposure of the Cr plating workers when compared with the other activities, which showed similar or higher values of inhalable and respirable air levels of total Cr and Cr(VI) (Table 2). However, as discussed above, Cr species are also likely to influence the difference between platers and welders, as seen in our regression analyses performed among those workers who did not use RPE. We speculated that the irregular use of RPE by platers might also lead to a higher hand-to-mouth contact influencing the overall exposure and also explain the higher levels of U-Cr obtained in this workers group. The reduced usage of RPE was also observed in a previous study conducted in the electroplating industry [2], where workers only used RPE when undertaking bulk additions to tanks or for some maintenance activities. The same was detected in a research project developed by the Health and Safety Executive in partnership with the Surface Engineering Association involving fifty-three companies engaged in nickel, Cr(VI) and/or cadmium electroplating. In this research project, RPE was not regularly used as a control measure for electroplating and only used at some specific sites when preparing bulk additions of solid chemicals to electroplate tanks or baths and related to certain maintenance activities where it was perceived that there was the potential for inhalation exposure [33].

The use of RPE should be considered as the last resort in the hierarchy of controls. Other preventive and protective measures should be considered first, e.g., elimination of the high-risk substance or substitution by a less toxic alternative, separating the substance from the workers, e.g., by automating the process or by use of engineering controls such as LEV. The effect of the latter technical control was shown to be effective and, as demonstrated in reduced U-Cr levels automatic as compared to manual Cr electroplating dipping, demonstrating that process automatisation is a viable option leading to a reduction in exposure. In some cases, however, the use of PPE such as RPE can be acceptable as a temporary measure for emergency work or during a temporary failure of controls where other means of control are not reasonably practicable.

RPE also needs to be appropriate for the agent and be well maintained and used correctly. Besides this, RPE only guarantees protection if not leaking; thus, it needs to be fit to the wearer’s face, and fit testing ensures that the equipment selected is suitable for the wearer. Preferably, this fit testing should be carried out by a competent person [39]. When the ‘in-use’ fit check of RPE is performed by the workers [39], the low frequencies obtained in all the activities, with most of the workers stating that this is not performed before starting activities that imply the use of RPE, combined with many questionnaires with missing information, may suggest that workers do not know what fit check or testing entails.

#### 4.3.2. Use of Gloves

Since the vast majority of workers were using gloves, the influence of gloves was only analysed for machining workers, with >10 workers (28.9%) not reporting the use of gloves. In workers who reported wearing gloves, this resulted in lower U-Cr. Gloves can undeniably protect from chemicals exposures avoiding hand contamination, if adequate to the chemical, but are also thought to promote exposure if not used correctly or changed frequently. The lower contamination levels found in hand wipe samples from the painters might be explained by the higher frequency of use of gloves when compared, for instance, with bath plating workers (Appendix A). Indeed, gloves can act as carriers of contamination and promote hands and surface cross-contamination [40,41]. The use of disposable gloves may reduce this cross-contamination [41] and, in our study, besides welders that use mostly welding gloves that are not disposable, in the other activities, the use of disposable gloves was the most often reported. However, chrome plating workers reported the use of reusable gloves for baths readjustment and Cr electroplating dipping (32.4% and 36.6%, respectively) (Appendix A). This is probably related to the type of gloves used, but, considering that these tasks are mostly undertaken manually, this option might promote surface and hand cross-contamination.

#### 4.3.3. Availability of LEV

Despite several missing answers concerning the presence of LEV in the workplaces, some inferences can be made. In plating, it seems that in most of the workplaces, LEV is present, as only 17% of the workers mentioned that this RMM was not available. However, when considering the availability of LEV by task, there were some workers reporting that some tasks involving proximity to the baths, particularly when not automatised, were performed without the use of LEV, such as baths readjustment and Cr electroplating dipping. Most of the workers stated that baths readjustment (72.7%), Cr electroplating dipping (79.8%) and sampling (100%) are still performed manually (Appendix A).

In welding (29.7%) and painting (47.1%), workers also reported the LEV to be available. In machining, this RMM was reported less frequently by workers (21.1%). This claims attention again for the need to apply more thoroughly the hierarchy of controls avoiding that worker’s protection is dependent solely on PPE.

In welding, the availability of LEV influenced the exposure by inhalation. The results for U-Cr and observed inhalable dust levels for both total Cr and Cr(VI) showed lower levels when LEV was in place (Table 4). Previously, the ventilation resources such as LEV were already reported as influencing exposure during welding operations [31,42]. A reduction in the median Cr(VI) concentrations in air by 68% was attributed to the availability of LEV in a study developed by Meeker and colleagues [42]. They also observed that the fraction of samples below the LOQ was higher for processes with LEV (61%) than for processes without LEV (49%). In our study, in terms of mean levels of total Cr and Cr(VI), the use of LEV corresponds to about a third of the air levels observed in the absence of LEV. Additionally, the existence of LEV also influenced the levels of total Cr in hand wipe samples for painting activity, with significantly lower levels in the presence of LEV. Previous reports also mentioned other variables as exposure determinants such as the type of welding process used, the material welded, degree of enclosure, worker’s age and welding experience and availability and use of PPE [30,31,43]. In our study, the years of experience in welding were positively and significantly correlated with pre- and post-shift levels of U-Cr, but for other variables, it was not possible to obtain detailed information due to missing answers, such as the composition of the material welded and degree of enclosure that might also influence the LEV effectiveness.

#### 4.3.4. Storage Working Clothes and RPE

A dedicated place for storing work clothes was related to higher levels of Cr in urine and hand wipe in painting activities (urine samples results). In plating activities, a statistically significant difference was seen despite the low number of workers in the “no” group. This finding may indicate that contaminated work clothes that are stored to be used on the next day may increase exposure by cross-contamination, e.g., by hand and mouth contact. Painting is normally an activity that implies high aerosolisation and consequent contamination of the working clothes and PPE that, if not washed after use or disposed of, can lead to secondary inhalation and dermal exposure. In a study developed previously by Beattie and colleagues [2], the existence of PPE lockers and the wearing of workwear at break times were considered the two mechanisms that explained the spread of contamination into clean areas such as canteens [2]. Even though the existence of a dedicated place for storing working clothes is a very positive measure since it avoids workers losing track of their PPE, it is important to be used to store only non-contaminated clothes or PPE. Therefore, clear procedures on when/which activities imply wash/dispose of the working clothes/PPE combined with more stringent housekeeping measures (e.g., frequent cleaning of storage places for working clothes and workplace surfaces) are fundamental to prevent exposure to Cr.

In bath plating, an opposite trend was observed for the availability of a dedicated storage location for RPE. This finding needs to be interpreted with caution because of the low number of “no” answers (Yes = 81, No = 5). However, this type of PPE needs a dedicated and clean space to be stored when not in use. These conditions need to be assured so RPE can act as a protection measure and not as a source of secondary contamination.

#### 4.3.5. Training on OSH Issues

The influence of previous training on OSH issues was only observed in welding, where reduced levels of Cr were observed in urine and hand wipes when training had occurred before. Indeed, in 2019 the European Commission suggested that welding fume emissions can often be reduced significantly (up to 50%) solely by the adjustment of welding parameters and that this can be accomplished by dedicated training of welders [44]. However, our results should be interpreted with caution because of the distribution of answers across categories, with a reduced number of workers reporting the absence of training (Yes = 175, No = 16). However, training on hazards and risks on used chemicals and on effective use of control measures can prevent exposure since workers themselves have a role in the prevention of exposure by following instructions on the use of PPE and personal hygiene [45].

#### 4.3.6. Previous Monitoring Actions

The high proportion of companies reporting on regulatory frameworks to be in place (OSH and REACH) probably explains why most of the workers (70.9%) who participated in the study were already involved in exposure monitoring campaigns (air monitoring and biomonitoring) aiming to evaluate worker’s exposure. This is particularly relevant in the case of welding and bath plating, where 100% and 75% of the workers, respectively, were from companies that developed previous air monitoring and biomonitoring campaigns. In bath plating, most of the exposure metrics (urine and air samples) resulted in lower levels when these campaigns occurred in the past. In welding, an opposite influence was observed. As reported in previous studies [2,3], if the purpose of the monitoring campaigns is exposure and risk assessment, then knowledge of exposure levels may contribute to higher quality risk assessment and communication, supporting the improvement of the RMMs and worker’s awareness that may lead to an exposure reduction. However, we did not collect information on how the data provided by the previous monitoring campaigns were used and if and how the results were communicated to the workers. Therefore, no firm conclusion can be made on the impact of monitoring campaigns. Future work will have to show the impact of this study on the participating companies.

### 4.4. Strenghts and Limitations

In this study, some improvements were identified to be considered in future multi-center occupational studies. For instance, most of the workers (80%) enrolled in the study were from companies that had already developed monitoring campaigns (air monitoring and biomonitoring). Therefore, some awareness of the need for these types of actions was already in place. This is a common source of bias in this type of occupational exposure study. Additionally, future studies might be relevant to collect information on the frequency of gloves and working clothes substitution and on the LEV maintenance regime since these aspects might influence the exposure levels. A major lesson learned is the value of collecting contextual data. Although the occupational hygiene dataset available in this study is considered a robust and valuable dataset, as mentioned in Galea et al. [18] and highlighted in the current manuscript, there were some deviations in the procedures for collection and analysis for occupational hygiene data. Analytical methods of the occupational hygiene samples were not included in the QC/QA programme within HBM4EU. The aim was that all air samples were to be analysed for both total Cr and Cr(VI). However, matching datasets for total Cr and Cr(VI) were provided in only two of the participating countries. This was related to the focus mainly on human biomonitoring in HBM4EU.

Despite the points mentioned here, and as mentioned in previous papers [16,17,18], the number of participants and samples collected from the nine involved countries allowed us to achieve the required statistical power for the study and obtain a more comprehensive and richer dataset covering a wide range of industrial applications of chromium and covering several chemical characteristics of this compound and related exposure conditions in a different type of companies across Europe. Additionally, this study covered many aspects of the working conditions and provided a rich dataset with quantified environmental and biological monitoring data at the individual level. Fieldwork also allowed us to engage workers’ companies and increase awareness of the need to control exposure to Cr(VI).

## 5. Conclusions

Some of the results from our study can be used to support recommendations and actions to be taken at the company and policy levels. The most important are shown below.

Collection of contextual data supports the interpretation of HBM and industrial hygiene data and the identification of exposure determinants;Both inhalation exposure and dermal exposure can be reflected as enhanced urinary chromium excretion;A high correlation between pre- and post-shift urinary chromium suggests that this biomarker reflects recent as well as past exposure;The relationship between inhalable air Cr(VI) levels and urinary Cr levels in platers are consistent with earlier reported regression equations published by Chen et al. [37];Similar inhalation exposures translate into two-fold higher U-Cr levels in chrome platers as compared to welders suggesting differences in toxicokinetic (e.g., absorption and bioavailability) of Cr(VI) compounds related to the route of exposure;In some specific chrome applications, the use of RPE contributes to reduced exposures and risk;Not all RMMs are equally effective in reducing exposure: automation of Cr electroplating dipping resulted in lower U-Cr levels; the use of RPE resulted in lower U-Cr in welding, bath plating and painting; LEV explains lower Cr exposure levels in welders;The existence of a dedicated place for storing working clothes might increase the exposure if not combined with clear procedures for washing/disposing of the working clothes/PPE and stringent housekeeping measures.Occupational health and safety training has a beneficial effect on exposure levels in welders.

The knowledge produced on which exposure determinants result in increased Cr exposure and the RMMs that are contributing more to control exposure allow future reflections at companies and policy levels, leading to safer working environments and public health protection.

## Figures and Tables

**Figure 1 ijerph-19-03683-f001:**
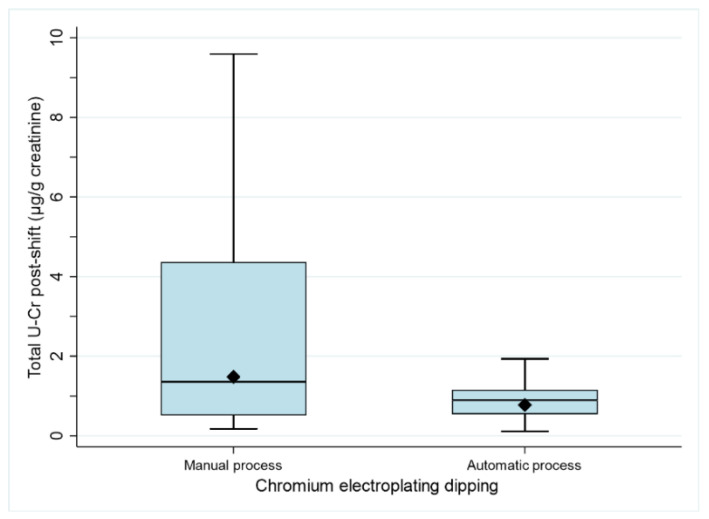
Levels of total urinary chromium (post-shift samples) for workers performing task “chromium electroplating dipping” by process type: manual (n = 67) or automatic n = 16). Box plots: The bottom and top of the box are, respectively, the 25th and 75th percentiles, and the horizontal line inside the box is the median (50th percentile). The lower and upper ends of the whiskers are the 5th and 95th percentiles, respectively. The solid diamond is the geometric mean.

**Figure 2 ijerph-19-03683-f002:**
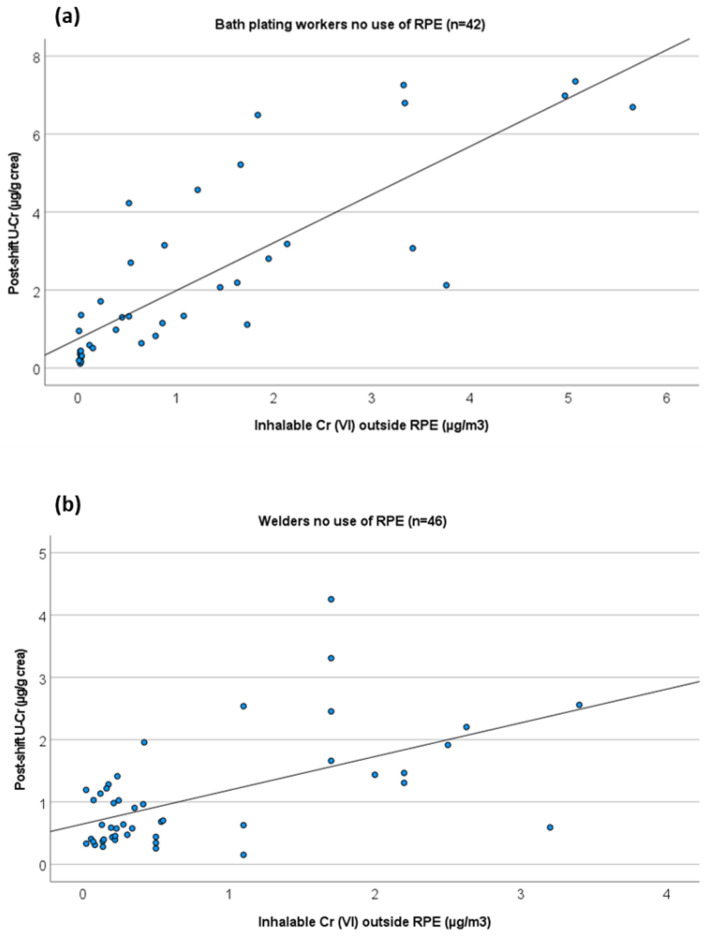
Regression analyses showing the relationship between U-Cr levels and inhalable Cr(VI) levels for (**a**) bath platers (y = 0.742 + 1.235x, r_s_ = 0.858, R^2^ = 0.679) and for (**b**) welders who did not use any RPE during the day of sampling (y = 0.647 + 0.541x, r_s_ = 0.515, R^2^ = 0.324).

**Table 1 ijerph-19-03683-t001:** Collection of information on exposure determinants by questionnaire.

Questionnaire	Determinants of Exposure
1	Previous monitoring campaigns (environment and biomonitoring)
	Previous training on OSH issues
2	Availability of LEV
	Use of gloves
	Use of RPE
	Daily fit check of RPE
	Existence of a dedicated place for storing working clothes and RPE
	Workers’ experience in their jobs
	Non-workplace exposure sources: smoking status, home location (urban or rural) and home traffic density

OSH—occupational safety and health; LEV—local exhaust ventilation; RPE—respiratory protection equipment.

**Table 2 ijerph-19-03683-t002:** Levels of total Cr and Cr(VI) in industrial hygiene samples (air and hand wipe samples).

		Total Cr (µg/m^3^) (Cr(VI) µg/m^3^)							
		n	Mean	GM	Median	P75	P95	Range	OEL Cr(VI) (µg/m^3^)
**Air samples** **(µg/m^3^)**	**Welding**								
	Inhalable—Outside RPE	124 (107)	111.0 (1.6)	18.9 (0.5)	16.1 (0.5)	73.3 (1.1)	481.0 (4.1)	0.2–3670.0 (<LOQ–40.4)	25.0 ^+^; 5.0 *
	Inhalable—Inside RPE	34 (10)	15.3 (1.6)	3.7 (1.0)	3.2 (0.5)	7.7 (1.1)	124.0 (4.1)	0.3–306.9 (0.1–44.3)	
	Respirable—Outside RPE	32 (20)	18.7 (2.3)	1.8 (0.2)	1.5 (0.1)	4.0 (1.2)	202.2 (22.3)	0.1–266.6 (0.2–22.8)	
	**Bath plating**								
	Inhalable—Outside RPE	31 (57)	41.1 (1.2)	7.7 (0.3)	9.9 (0.4)	32.4 (1.7)	359.0 (5.1)	0.1–621.3 (<LOQ–9.1)	10.0 ^+^; 5.0 *
	Respirable—Outside RPE	34 (54)	7.8 (0.4)	1.1 (0.1)	0.7 (0.1)	2.7 (0.5)	59.3 (2.3)	0.9–166.3 (<LOQ–3.1)	
	**Painting**								
	Inhalable—Outside RPE	4 (7)	30.3 (29.3)	7.6 (5.8)	19.1 (5.6)	70.8 (154)	82.0 (154)	1.0–82 (0.6–154.4)	
	Respirable—Outside RPE	11 (<LOQ)	2.5	1.2	1.0	3.0	9.5	0.3–9.4	
	**Machining**								
	Inhalable—Outside RPE	8 (15)	42.0 (0.2)	11.0 (0.1)	48.7 (0.1)	70.2 (0.2)	96.3 (<LOQ)	0.3–96.3 (<LOQ–0.4)	
	Respirable—Outside RPE	9 (10)	1.5 (0.03)	0.7 (0.03)	0.6 (0.03)	2.5 (0.04)	6.2 (0.05)	0.2–6.2 (<LOQ–0.05)	
	**Steel production**								
	Inhalable—Outside RPE	5 (<LOQ)	4.9	3.3	2.4	9.6	13.9	1.5–13.9	
	Respirable—Outside RPE	5 (<LOQ)	0.9	0.6	0.3	1.9	2.0	0.3–2.0	
	**Maintenance and laboratory work**								
	Inhalable—Outside RPE	1 (3)	<LOQ	9.9 (0.4)	<LOQ	<LOQ	<LOQ	9.9 (0.3–0.8)	
	Respirable—Outside RPE	2 (2)	0.4 (0.2)	0.4 (0.1)	<LOQ	<LOQ	<LOQ	0.2–0.6 (<LOQ–0.3)	
	**Thermal spraying**								
	Inhalable—Outside RPE	5 (5)	2566 (12.5)	1050 (11.4)	823 (9.6)	5755 (18.8)	8359 (21.0)	192.5–8359.5 (6.4–21.0)	
	Respirable—Outside RPE	5 (5)	58.6 (0.07)	23.0 (0.06)	9.6 (0.06)	136.0 (0.1)	140.0 (0.1)	5.5–140.0 (<LOQ–0.1)	
**Wipe samples** **Shift sum **** **(µg/cm^2^)**	**Welding**	115	0.3	0.1	0.2	0.3	1.0	<LOQ–1.8	NA
	**Bath plating**	77	0.6	0.1	0.1	0.7	2.3	<LOQ–8.4	
	**Painting**	32	0.1	0.1	0.0	0.1	0.3	<LOQ–0.3	
	**Machining**	25	0.2	0.1	0.1	0.1	1.3	<LOQ–1.4	
	**Steel production**	5	<LOQ	<LOQ	<LOQ	<LOQ	<LOQ	<LOQ	
	**Maintenance and laboratory work**	8	0.1	<LOQ	<LOQ	0.3	0.5	<LOQ–0.5	
	**Thermal spraying**	5	18.5	13.9	13.8	32.7	46.6	6.6–46.6	

LOQ = Limit of quantification; GM = Geometric mean; RPE = respiratory protection equipment; P75 = Percentile 75; P95 = Percentile 95; OEL = Occupational Exposure Limit; NA = Not available; ^+^ actual OEL; * stricter OEL to be applied in January 2025; ** Sum of the samples taken during the shift and post-shift was calculated and presented as a “shift sum”.

**Table 3 ijerph-19-03683-t003:** Urinary concentrations of total Cr in the population of workers studied and compared with controls. Results are adjusted for creatinine.

			Total U-Cr (µg/g Creatinine)						Correlation Pre-Shift vs. Post-Shift
		n	Mean	GM	Median	P75	P95	Range	r_s_
**Pre-shift**	**Workers** ^a,b,c,d^	399	0.9	0.6	0.5	1.0	3.1	0.0–8.3	0.795
	Welding ^a,b,c,d^	193	0.7	0.5	0.5	0.9	2.0	0.1–5.8	0.797
	Bath plating ^a,b,c,d^	90	1.5	0.8	0.8	2.4	5.0	0.1–8.3	0.892
	Painting ^a,b,c,d^	52	0.8	0.5	0.6	1.2	2.6	0.1–4.0	0.703
	Machining ^a,b,c,d^	38	0.7	0.5	0.4	1.0	1.9	0.1–2.9	0.588
	Steel production	11	0.9	0.6	0.5	1.0	4.5	0.2–4.4	-
	Maintenance and laboratory work	8	0.4	0.3	0.2	0.7	1.1	0.1–1.1	-
	Thermal spraying	5	0.6	0.4	0.4	1.3	2.1	0.1–2.1	-
	**Controls ***	135	0.4	0.2	0.2	0.4	1.3	0.0–3.2	-
	Within company controls ^e^	94	0.4	0.3	0.3	0.5	1.4	0.0–3.2	-
	Outwith company controls ^e^	41	0.2	0.1	0.1	0.2	0.4	0.1–1.9	-
**Post-shift**	**Workers** ^a,b,d^	399	1.4	0.8	0.9	1.7	5.1	0.1–13.6	-
	Welding ^d,e^	189	1.1	0.7	0.7	1.4	3.4	0.1–5.8	-
	Bath plating ^a,b,d^	90	2.3	1.2	1.1	2.4	7.7	0.1–13.6	-
	Painting ^d^	45	1.4	0.7	0.9	1.8	3.6	0.1–12.3	-
	Machining ^a,b,d^	36	1.5	1.0	1.0	1.7	6.7	0.1–7.7	-
	Steel production	10	1.2	1.1	1.3	1.7	2.0	0.3–2.0	-
	Maintenance and laboratory work	8	0.5	0.4	0.3	1.0	1.5	0.1–1.5	-
	Thermal spraying	5	0.7	0.4	0.4	1.4	2.4	0.1–2.4	-

^a^ Statistically significant differences (*p* < 0.05) between workers and within company controls; ^b^ Statistically significant differences (*p* < 0.05) between workers and outwith company controls; ^c^ Statistically significant differences (*p* < 0.05) between workers and all controls; ^d^ Statistically significant differences (*p* < 0.05) between pre-shift and post-shift; ^e^ Statistically significant differences between within company and outwith company controls; * Controls used in [17] and the information was adapted with permission from [17]; P75 = Percentile 75; P95 = Percentile 95; r_s_ = Spearman correlation coefficient.

**Table 4 ijerph-19-03683-t004:** Heatmap representing Spearman coefficient correlation (r_s_) analysis for total Cr and Cr(VI) levels in biological samples (urine) and industrial hygiene samples (air and wipe). Only significant correlations are displayed (Sig. = *p* < 0.05). Red cell: ≤0.2 = poor; Orange cell: 0.2 ≤ 0.5 = fair; Light green: 0.5 ≤ 0.7 = moderate; Dark green: 0.7 ≤ 0.9 = very strong; Grey cell: tested, but no significant correlation found.

		Urine: Total Cr (Post Shift) (µg/g Creatinine)	Air: Inhalable Total Cr Outside RPE (µg/m^3^)	Air: Inhalable Total Cr Inside RPE (µg/m^3^)	Air: Inhalable Cr(VI) Outside RPE (µg/m^3^)	Air: Inhalable Cr(VI) Inside RPE (µg/m^3^)	Air: Respirable Total Cr Outside RPE (µg/m^3^)	Air: Respirable Cr(VI) Outside RPE (µg/m^3^)	Wipe: Total Cr Shift Sum (µg/m^2^)
Urine: Total Cr (Pre-shift) (µg/g creatinine)	r_s_	0.795	0.165		0.476	0.412	0.369	0.677	0.394
	Sig.	<0.001	0.047		<0.001	0.005	<0.001	<0.001	<0.001
	N	382	145		161	44	98	91	266
Urine: Total Cr (End shift) (µg/g creatinine)	r_s_				0.461	0.514	0.329	0.694	0.403
	Sig.				<0.001	<0.001	0.001	<0.001	0.001
	N				193	44	96	90	260
Air: Inhalable Total Cr Outside RPE (µg/m^3^)	r_s_				0.609		0.800	0.457	0.606
	Sig.				<0.001		<0.001	0.005	<0.001
	N				88		84	36	90
Air: Inhalable Total Cr Inside RPE (µg/m^3^)	r_s_					0.435			
	Sig.					0.007			
	N					37			
Air: Inhalable Cr(VI) Outside RPE (µg/m^3^)	r_s_						0.654	0.791	0.495
	Sig.						<0.001	<0.001	<0.001
	N						48	88	143
Air: Inhalable Cr(VI) Inside RPE (µg/m^3^)	r_s_								
	Sig.								
	N								
Air: Respirable Total Cr Outside RPE (µg/m^3^)	r_s_							0.587	0.479
	Sig.							<0.001	<0.001
	N							34	97
Air: Respirable Cr(VI) Outside RPE (µg/m^3^)	r_s_								0.639
	Sig.								<0.001
	N								91

**Table 5 ijerph-19-03683-t005:** Exposure determinants and their effect on workers’ exposure levels (Mann–Whitney test and Kruskal–Wallis test).

Activity	RMM	Urine Total Cr	Air Inha out-RPE Total Cr	Air Inha out-RPE Cr(VI)	Air Inha in-RPE Cr(VI)	Air Resp out-RPE Cr(VI)	Hand Wipe Total Cr
**Welding** **(n = 195)**	Use of RPE	Yes (*p* = 0.004)	------	------	------	------	------
	Daily fit check of RPE	No	------	------	------	------	------
	Use of gloves	No	------	------	------	------	No
	Availability of LEV	Yes (*p* = 0.001)	Yes (*p* = 0.015)	Yes (*p* < 0.001)	No	No	No
	Dedicated place for storing work clothes	No	------	------	------	------	No
	Dedicated place for storing RPE	No	------	------	------	------	No
	Previous training	Yes (*p* = 0.010)	------	------	------	------	Yes (*p* = 0.005)
	Previous monitoring campaigns	Yes (*p* < 0.001) ^a^	No	Yes (*p* < 0.001) ^b^	Yes (*p* = 0.001) ^b^	No	No
**Bath** **plating** **(n = 90)**	Use of RPE	Yes (*p* = 0.002)	------	------	------	------	------
	Daily fit check of RPE	No	------	------	------	------	------
	Use of gloves	No	------	------	------	------	No
	Availability of LEV	No	No	No	ND	No	No
	Dedicated place for storing work clothes	Yes (*p* = 0.008) **	------	------	------	------	Yes (*p* = 0.024) **
	Dedicated place for storing RPE	Yes (*p* = 0.013)	------	------	------	------	Yes (*p* = 0.002)
	Previous training	No	------	------	------	------	No
	Previous monitoring campaigns	Yes (*p* < 0.001) ^b^	No	Yes (*p* = 0.014) ^b^	ND	Yes (*p* = 0.014) ^b^	Yes (*p* = 0.014) ^b^
**Painting** **(n = 52)**	Use of RPE	Yes (*p* = 0.026)	------	------	------	------	------
	Daily fit check of RPE	No	------	------	------	------	*
	Use of gloves	No	------	------	------	------	No
	Availability of LEV	No	*	*	ND	*	Yes (*p* = 0.022)
	Dedicated place for storing work clothes	Yes (*p* = 0.007) **	------	------	------	------	No
	Dedicated place for storing RPE	No	------	------	------	------	No
	Previous training	*	------	------	------	------	*
	Previous monitoring campaigns	*	*	*	ND	*	*
**Machining** **(n = 38)**	Use of RPE	No	------	------	------	------	------
	Daily fit check of RPE	No	------	------	------	------	------
	Use of gloves	Yes (*p* = 0.003)	------	------	------	------	No
	Availability of LEV	No	No	No	ND	No	No
	Dedicated place for storing work clothes	No	------	------	------	------	No
	Dedicated place for storing RPE	No	------	------	------	------	No
	Previous training	No	------	------	------	------	No
	Previous monitoring campaigns	*	*	*	ND	*	*

RMM = Risk Management Measure; RPE = Respiratory Protective Equipment; LEV = Local Exhaust Ventilation; Air Inha out-RPE = air inhalable outside RPE; Air Inha in-RPE = air inhalable inside RPE; Air Resp out-RPE = air respirable outside RPE; ND—not determined—measurements of total Cr and Cr(VI) inside RPE air samples were performed only for welders; * unable to compute due to low number of observations or absence of data; ** Higher values when a dedicated place for storing work clothes is present; ^a^ Statistically significant difference (*p* < 0.05) between categories “environmental”, “none” and “both”. ^b^ Statistically significant difference (*p* < 0.05) between categories “none” and “both”; Dashed line: statistical analysis not performed.

## Data Availability

Not applicable.

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
