# Peer review of "HBM4EU Chromates Study: Determinants of Exposure to Hexavalent Chromium in Plating, Welding and Other Occupational Settings"

_ijerph, 2022, doi:10.3390/ijerph19063683_

Round 1

Reviewer 1 Report

I am very grateful for the opportunity to review this interesting and well written paper that aims to analyze and evaluate Cr exposure and assess the influence of exposure control of implemented RMMs. This is a work of high scientific interest with valuable contributions. TTherefore, the quality of this article is adequate to be published in IJERPH. However, the authors should take into account some advice to improve it:

Line 64-65 : The text says: "The use of Cr(VI) compounds (chromates, chromium trioxide and dichromium tris(chromate)) is authorized under the Registration, Evaluation, Authorization and Restriction of Chemicals (REACH) regulation". Authors should clarify that this regulation is a European Union regulation, not aplicable to other countries.

Methods section: This section would be notably improved if the authors would briefly describe the laboratory methods used for Cr analysis for air urine and dermal wipe samples.

Line 516 : The authors claim: "No influence of tobacco smoke was detected in exposed workers or in controls". How do you control that?

Author Response

I am very grateful for the opportunity to review this interesting and well written paper that aims to analyze and evaluate Cr exposure and assess the influence of exposure control of implemented RMMs. This is a work of high scientific interest with valuable contributions. Therefore, the quality of this article is adequate to be published in IJERPH. However, the authors should take into account some advice to improve it:

Author’s response: Thank you for the positive comment and valuable suggestions to improve our manuscript.

Line 64-65: The text says: "The use of Cr(VI) compounds (chromates, chromium trioxide and dichromium tris(chromate)) is authorized under the Registration, Evaluation, Authorization and Restriction of Chemicals (REACH) regulation". Authors should clarify that this regulation is a European Union regulation, not applicable to other countries.

Author’s response: It was clarified in the text. Added the following sentence: In the European Union, the use of Cr(VI) compounds (chromates, chromium trioxide and dichromium tris(chromate)) is authorized under the Registration, Evaluation, Authorization and Restriction of Chemicals (REACH) regulation. REACH was adopted to improve the protection of human health and the environment from the risks that can be posed by chemicals, while enhancing the competitiveness of the EU chemicals industry.

Methods section: This section would be notably improved if the authors would briefly describe the laboratory methods used for Cr analysis for air urine and dermal wipe samples.

Author’s response: With due consideration of the comments expressed by Reviewers 2 and 3 we do not wish to further increase the length of our manuscript. The analytical methods performed for each type of samples has already been described in previous publications. As such, we provide the following clarification in section 2.2.: The samples analyses were performed as described in previous publications by the HBM4EU chromates study [16, 17].

Line 516 : The authors claim: "No influence of tobacco smoke was detected in exposed workers or in controls". How do you control that?

Author’s response: It was possible to conclude this with a dedicated statistical analysis, where we compared the U–Cr levels between smokers and non-smokers in workers and controls. No statistically significant differences were found between levels of U-Cr for smokers and non-smokers. Besides line 515, these data were referred also in lines 486-490 (Results section).

Reviewer 2 Report

The manuscript addresses a significant public health issue (occupational safety and health).
According to an estimate by Cherrie et al. (British Journal of Cancer, 2017), there will be 10 000 new
occupational cancer deaths from Chrormate in the next 60 years. The study is well planned and the
manuscript is informed and well written. The conclusions are comprehensible to the reader. Whether
the manuscript needs to be shortened should be decided by the Editor in Chief.

Author Response

The manuscript addresses a significant public health issue (occupational safety and health). According to an estimate by Cherrie et al. (British Journal of Cancer, 2017), there will be 10 000 new occupational cancer deaths from Chromate in the next 60 years. The study is well planned and the manuscript is informed and well written. The conclusions are comprehensible to the reader. Whether the manuscript needs to be shortened should be decided by the Editor in Chief.

Author’s response: Thank you for the positive comments.

Reviewer 3 Report

In abstract, LEV should be spelt completely. Is there any other observations or citations regarding to the effects of LEV?

Different sampling approaches were used for inhalable dusts; however, is it suitable for comparing the data together? Is there any study by authors on the variations of different samplers for dust?

Tables 2 and 3 are large and many data are presented. The original data should be shifted to the Supplement Section. Authors should imply important data in figures or summarize the data in a simplified table.

Original Tables 4 and 5 should be shifted to the Supplement Section. Authors should simplify tables.

Original Table 7 should be simplified. It is too complex for reading.

Author Response

In abstract, LEV should be spelt completely. Is there any other observations or citations regarding to the effects of LEV?

Author’s response: We made the requested change in the abstract, thank you for spotting this.  Within our manuscript we already refer to previous publications where the influence of LEV is reported.  For example, we kindly draw the reviewers’ attention to the examples cited in workers exposure section (4.1) and availability section (4.3.3), with the relevant citations being detailed below:

Persoons, R.; Arnoux, D.; Monssu, T.; Culié, O.; Roche, G.; Duffaud, B.; Chalaye, D.; Maitre, A. Determinants of occupational exposure to metals by gas metal arc welding and risk management measures: a biomonitoring study. Toxicol Lett. 2014, 231, 135-41. doi: 10.1016/j.toxlet.2014.09.008.

Lehnert, M.; Weiss, T.; Pesch, B.; Lotz, A.; Zilch-Schöneweis, S.; Heinze, E.; Van Gelder, R.; Hahn, J.U.; Brüning, T. Reduction in welding fume and metal exposure of stainless steel welders: an example from the WELDOX study. Int Arch Occup Environ Health. 2014; 87, 483-92.

Meeker, J.D.; Susi, P.; Flynn, M.R. Hexavalent chromium exposure and control in welding tasks. J Occup Environ Hyg. 2010, 7, 607-15.

IARC. Welding, Molybdenum Trioxide, and Indium Tin Oxide. Retrieved from. https://publications.iarc.fr/Book-And-Report-Series/Iarc-Monographs-On-The-Identification-Of-Carcinogenic-Hazards-To-Humans/Welding-Molybdenum-Trioxide-And-Indium-Tin-Oxide-2018

Different sampling approaches were used for inhalable dusts; however, is it suitable for comparing the data together? Is there any study by authors on the variations of different samplers for dust?

Author’s response: The reviewer states that different samplers were used for the inhalable dusts. In our study the personal inhalable dust fraction was collected in the breathing zone using an IOM sampling head (flow rate 2 L/min). When sampling this fraction for the welders, the SKC Mini-sampler was used in the UK, Belgium and Luxembourg, with the IOM head being used in all other countries. All of these sampling devices adhere to CEN-EN 481:1993 Workplace atmospheres - Size fraction definitions for measurement of airborne particles. Goran et al (2009) reports that the sampling bias of the mini sampler versus the IOM sampler depends on the size distribution of the sampled aerosol and that for manganese (Chromium was not assessed) the negative root mean square sampling bias of the mini sampler versus the IOM sampler is −0.046 which was statistically non-significant. We did not assess the size distribution of the aerosols at each of the sites and were unable to establish from the literature if any suitable correction factors were needed and could be sensibly applied for total Cr for welding aerosols. As such the data has been pooled. 

Göran Lidén, Jouni Surakka, A Headset-Mounted Mini Sampler for Measuring Exposure to Welding Aerosol in the Breathing Zone, The Annals of Occupational Hygiene, Volume 53, Issue 2, March 2009, Pages 99–116, https://doi.org/10.1093/annhyg/mep001

Tables 2 and 3 are large and many data are presented. The original data should be shifted to the Supplement Section. Authors should imply important data in figures or summarize the data in a simplified table. Original Tables 4 and 5 should be shifted to the Supplement Section. Authors should simplify tables. Original Table 7 should be simplified. It is too complex for reading.

Author’s response: We moved Tables 2 and 3 into the supplementary data section of the manuscript (renumbering the remaining tables accordingly) and added some text concerning some relevant information available in table 3 (lines 303-305).

We made some changes in Table 7 to be easier to read. The description of samples was also abbreviated and described in the table footnote. The following columns were removed:

  • column “answers” – this column contained the number of answers by variable categories. Data was inserted in the text where needed to support the results obtained (line 485, line 791, line 801).
  • Columns “air inhalable inside RPE Total Cr” and “air respirable outside RPE Total Cr” – results obtained from Mann-Whitney and Kruskal-Wallis tests revealed no association between levels of total Cr and RMM implemented. Therefore, the columns were removed, and these results were added to the text (lines 486-488) in the sentenceThe statistical analysis did not reveal any association between the RMM and the levels of total Cr in inhalable inside RPE and respirable outside RPE air samples (data not shown).”

Round 2

Reviewer 3 Report

 Thanks for authors' revisions for the comments.